# Metabolic Profiling in Tuberous Roots of *Ranunculus asiaticus* L. as Influenced by Vernalization Procedure

**DOI:** 10.3390/plants12183255

**Published:** 2023-09-13

**Authors:** Giovanna Marta Fusco, Petronia Carillo, Rosalinda Nicastro, Letizia Pagliaro, Stefania De Pascale, Roberta Paradiso

**Affiliations:** 1Department of Environmental, Biological and Pharmaceutical Sciences and Technologies, University of Campania Luigi Vanvitelli, 81100 Caserta, Italy; giovannamarta.fusco@unicampania.it (G.M.F.); rosalinda.nicastro@unicampania.it (R.N.); letizia.pagliaro@studenti.unicampania.it (L.P.); 2Department of Agricultural Sciences, University of Naples Federico II, 80055 Portici, Italy; depascal@unina.it

**Keywords:** geophytes, tuberous roots, cold requirement, GABA, BCAAs, polyphenols

## Abstract

*Ranunculus asiaticus* L. is an ornamental geophyte. In commercial practice, it is mainly propagated by rehydrated tuberous roots. Vernalization before planting is a common practice to overcome the natural dormancy of tuberous roots; however, little is known about the mechanisms underlying the plant’s response to low temperatures. We investigated the influence of three preparation procedures of tuberous roots, only rehydration (control, C), and rehydration plus vernalization at 3.5 °C for 2 weeks (V2) and for 4 weeks (V4), on plant growth, leaf photosynthesis, flowering, and metabolism in plants of two hybrids, MBO (early flowering, pale orange flower) and MDR (medium earliness, bright orange flower), grown in pots in an unheated greenhouse. We reported the responses observed in the aerial part in a previous article in this journal. In this paper, we show changes in the underground organs in carbohydrate, amino acids, polyphenols, and protein levels throughout the growing cycle in the different plant stages: pre-planting, vegetative growth, and flowering. The metabolic profile revealed that the two hybrids had different responses to the root preparation procedure. In particular, MBO synthesized GABA and alanine after 2 weeks and sucrose after 4 weeks of vernalization. In contrast, MDR was more sensitive to vernalization; in fact, a higher synthesis of polyphenols was observed. However, both hybrids synthesized metabolites that could withstand exposure to low temperatures.

## 1. Introduction

*Ranunculus asiaticus* L. (Family Ranunculaceae) is a perennial geophyte cultivated for cut flower and flowering potted plant production that is gaining attention for gardening and landscape design [1]. The cultivation of *R. asiaticus* has been growing in recent years, thanks to the advance of knowledge on vegetative propagation through underground storage organs (namely tuberous roots) and the development of new hybrids [1]. As with several spring-flowering geophytes, *R. asiaticus* exhibits a summer rest period and requires a warm-cold-warm sequence to complete its life cycle [1]. In fact, in the cool and wet winter period, plants are in the active growth stage and flowering, while in the dry and hot summer, they require a quiescence phase [2]. In the Mediterranean climate, in the wild, the first autumn rainfalls rehydrate the tissues of the dry, dormant tuberous roots of spontaneous *ranunculus,* making them sprout and develop leaf rosettes [3]. Flowering lasts from February to May, when the aerial part wilts and the roots enter dormancy [1]. Unlike other geophytes, in *R. asiaticus,* the shoot apical meristem of dormant tuberous roots is inactive, while its activity restarts under mild temperatures after rehydration. Tuberous roots may survive long quiescence, tolerating prolonged storage, so that *ranunculus* is recognized as a “resurrection geophyte” species [4,5]. During root growth, cortical cell size increases and cell walls become rich in pectin materials, while starch granules and protein bodies accumulate for the next growth phase of plants upon awakening from dormancy [4,5]. Tuberous roots exhibit an annual cycle: old roots store nutrients and die after hydration and nutrient transport to the aerial part, and new roots develop during the growth stage and enter dormancy in the summer [3].

In *R. asiaticus*, flowering and tuberization are in antagonism, and both are influenced by the thermo- and photo-periods. Short days and low temperatures promote meristematic growth, increasing the number of buds, leaves, and flowers [1], while long days anticipate the flowering of preformed buds, reducing the flower yield and quality while fostering the enlargement of tuberous roots [1,6]. Flowering earliness, flower yield, and flower dimensions vary with the plant genotype, the size, and the storing and preparation procedure of tuberous roots before planting, as well as the plant’s growing conditions [1,7].

Plant cultivation of *R. asiaticus* for commercial production can start either via seeds or via tuberous roots, harvested at the end of the growing period and prepared as propagation material through natural dehydration to less than 15% moisture content. However, planting tuberous roots anticipates the beginning of flowering and increases the flower yield compared to sowing seeds [1,3]. However, flowering is also influenced by the thermal history of roots before and after planting and the photoperiod during plant growth, as plants show a low temperature requirement (5–10 °C in the night) and a quantitative (or facultative) photoperiodic response to long days (LD) [3]. Similarly to low winter temperatures in nature, cold treatments of tuberous roots (vernalization) accelerate sprouting, leaf rosette formation, and flowering and increase the number of flowers compared to only rehydration [4], promoting the release from summer dormancy [5]. Indeed, low temperatures induce starch breakdown [8], increasing the availability of free sugars like sucrose [9], down-regulating the biosynthesis of abscisic acid (ABA), and promoting that of active forms of gibberellic acid (GA) [10].

In geophytes, the dormant organs are enriched in carbohydrates, mainly starch, which are broken down when the vernalization ends, allowing the accumulation of oligosaccharides or monosaccharides-free sugars (e.g., sucrose and hexoses) [9]. In potatoes, the dormancy release and the following starch breakdown exert an activating action on glycolysis and pentose phosphate pathways, providing carbon skeletons and energy for faster tuberous sprouting [10]. This process could also provide both cryoprotection for the developing leaf rosette and an energy source for the following bolting process [6]. Indeed, GA has an antagonistic role against ABA in the regulation of bulb dormancy [11]. The exposure to low temperatures allows the overcoming of dormancy by repressing and/or downregulating the genes involved in the ABA and jasmonic acid (JA) biosynthetic pathways, thus relieving the antagonistic effect of ABA and JA on GA signaling and promoting sprouting [11,12]. Vernalization can decrease endogenous ABA in potato tubers; similarly, vernalization as well as cutting treatment inhibit the expression of the enzyme 9-cis-epoxycarotenoid dioxygenase (NCED), which catalyzes the rate-limiting step in ABA biosynthesis, thus reducing ABA content [10,13].

Vernalization of tuberous roots is a common practice to schedule cut flower production of *R. asiaticus*, as in many ornamental geophytes [2]. However, it is known that the response to low temperatures is genotype-dependent and changes among the hybrids, and data on the effects of cold treatments on the primary and secondary metabolites and the mechanisms underlying the plant response to vernalization and, more generally, about plant physiology and metabolism in this species are quite scarce.

As reported in Carillo et al. [14], in geophytes, the tuberous roots act as storage organs, and in particular, they synthetize starch. Sometimes in the roots, this sugar is combined with other sugar-based polymers such as glucomannan and fructans that can help or replace starch [15] to overcome prolonged adverse environmental conditions. In addition, small amounts of storage proteins in the tuberous root can be associated with overwintering [4]. In fact, stress-related proteins (dehydrins and smHSPs) and putative storage proteins, in particular the 14 kDa protein (Ra14), which is the major *R. asiaticus* root protein, increase during desiccation, with their expression positively modulated by dehydration and ABA [4,16]. These proteins are rapidly convertible to free amino acids at the end of dormancy, which is useful not only to promote fast growth but also to help the plant overcome the oxidative stress effects due to vernalization (e.g., proline and GABA). In fact, in the tuberous roots of *R. asiaticus,* the fast starch hydrolysis upon vernalization promotes not only the synthesis and accumulation of sucrose but also of other primary metabolites acting as osmolytes and antioxidants, like proline, alanine, and γ-aminobutyric acid (GABA) [17,18]. In addition, minor amino acids, and particularly branched chain amino acids (BCAAs), whose synthesis increases when hexoses increase under stress [19], may exert a scavenging function against reactive oxygen species (ROS) or an anaplerotic function by supplying intermediates to the mitochondrial electron transport chain, supporting plant metabolism and flowering [18,20,21].

Due to the importance of tuberous roots as a reserve organ in living plants and propagation material in commercial floriculture, it is important to gain new knowledge on their metabolic changes, focusing on the source-sink relationships with the plant’s aerial part from the dormancy period to sprouting and flowering [4].

We carried out a series of experiments aiming at elucidating the influence of temperature [17,18] and photoperiod [14,15,16,17,18,19,20,21,22,23] and their interaction on plant physiology and metabolism and the flowering process of *R. asiaticus*. In the present experiment, we investigated the influence of three preparation procedures of tuberous roots—only rehydration and two times of exposure to cold temperatures after rehydration—on plant growth and metabolic profiles in two hybrids with different flowering earliness grown in pots in an unheated glasshouse. We reported results on the responses observed in the aerial part in a previous article in this journal [18]. In the present paper, we show data on the metabolism of tuberous roots that demonstrate a different response of the two hybrids depending on the genotype and sensitivity to two different vernalization procedures. We focused the targeted metabolic profiling on specific classes of metabolites, and, in particular, those of carbohydrates and amino acids, because these are preferably synthesized by plants in response to cold/oxidative stress both for their protective function, as mentioned above, but are also easily recycled for the anaplerotic provision of carbon skeletons and energy for the growth of plants upon stress relief [17,18,19,20,21]. Instead, polyphenols were chosen as a representative class of metabolites that, despite being very effective as antioxidants, have a high cost in energy terms, both for their synthesis and for their non-easy recyclability, so as to slow down the growth. This targeted metabolic profiling analysis in both storage organs and leaves contributes to unveiling the strategies adopted by the different hybrids to overcome the cold/oxidative stress deriving from vernalization, in addition to being useful in the production scheduling of *R. asiaticus* L.

## 2. Results

### 2.1. Tuberous Roots’ Metabolic Profile

The hybrid (H), vernalization (V), and interaction “*Hybrid × Vernalization procedure*” (H × V) significantly affected the metabolites in the two hybrids of *Ranunculus asiaticus* L. MDR and MBO in the different growth stages (pre-planting, vegetative growth, and flowering). In particular, the content of glucose was significantly influenced by H in pre-planting and flowering, being higher in MDR than in MBO (+19.3% and +113%, respectively) (Table 1). This parameter was also affected by H × V in all growth phases. In vegetative growth, the glucose content underwent a drastic decrease in MDR V4 (−51%) and in MBO V2 (−54%) compared to the respective controls (Table 1); while at flowering, it increased in MBO V2 (+54.5%) and decreased in both hybrids under V4 (−47.8% and −92.9%, respectively), compared to the respective controls (Table 1).

The fructose content was influenced by both V and H × V in the vegetative stage and only by H × V at flowering. Specifically, in the vegetative phase, fructose significantly increased only in MDR V4 compared to the other two MDR treatments (+70% on average). In this stage, the duration of the vernalization procedure (V) determined contrasting effects in MBO, decreasing fructose by 66% in MBO V2 and increasing it by 239% in MBO V4 compared to control (Table 1). On the contrary, at flowering, fructose decreased in MBO V4 (−96%) compared to control.

In pre-planting, the sucrose content was slightly higher in MDR than in MBO (on avg. 8.63 mg g^−1^ DW and 8.28 mg g^−1^ DW, respectively). Instead, sucrose was significantly affected by H × V in vegetative and flowering phases, and also by V in vegetative phase only. Particularly in the former phase, sucrose increased in both MDR V4 and MBO V4 (+55% and +271%, respectively), while decreasing only in MBO V2 (−77%), compared to respective controls (Table 1). At flowering, only MBO V4 significantly decreased compared to control (−96%).

Starch was affected by H in the pre-planting and flowering stages. In fact, in both of these stages, this parameter was lower in MDR (on avg. 39.55 and 38.42 mg g^−1^ DW, respectively) than in MBO (on avg. 52.86 and 53.75 mg g^−1^ DW). The starch content was also affected by H × V in all growth stages (Table 1).

The polyphenol content was affected by H, V, and H × V. In particular, in vegetative growth and flowering, it was higher in MDR (on avg. 12.64 and 10.13 µg mg^−1^ DW, respectively) than in MBO (on avg. 8.01 and 6.01 µg mg^−1^ DW, respectively). In the vegetative growth, polyphenols increased in MDR V2 (+15%) while decreasing in MDR V4 (−43%) compared to control. At flowering, polyphenols increased in MDR V2 and V4 (+119% and 99.8%, respectively) but decreased in MBO V2 and V4 (−56% and −77%, respectively) compared to their respective controls (Table 1).

Alanine, GABA, glycine, and proline were the amino acids undergoing the main changes in vernalized MDR in the pre-planting stage. In fact, these amino acids increased in MDR V4 (+387%, 572%, 59%, and 78%, respectively) and also GABA in V2 (+173%) compared to the respective controls (Figure 1A,D,G and Figure 2B). In the same hybrid, there was a decrease in the content of arginine in MDR V2 (−46%; Figure 1B), glutamine in MDR V4 (−42%; Figure 1F), and threonine in both vernalization procedures V2 and V4 (−24% and −20%, respectively; Figure 2D) compared to their controls. Some of these amino acids also decrease during vegetative growth and flowering. Specifically, in both stages, GABA and glutamate decreased in MDR V2 (−67% and −62%, respectively) and MDR V4 (−88% and −35%, respectively) in the vegetative stage, while at flowering, GABA decreased in MDR V2 (−74%) and in MDR V2 (−58%), and glutamate in MDR V2 and V4 (−61%), compared to controls (Figure 1D,E).

In the hybrid MDR, in the vegetative stage, asparagine, proline, and threonine decreased in MDR V2 (−325%, −44%, and −83%, respectively), and asparagine also decreased in MDR V4 (−237%) compared to control (Figure 1B and Figure 2B,D). At flowering, alanine decreased in MDR V4 (-50%; Figure 1B), and MEA in both procedures V2 and V4 (−61% and −58%, respectively; Figure 1H). Conversely, in MDR V2, glutamine increased compared to control (+111%) (Figure 1F). Regarding the hybrid MBO, in pre-planting, alanine, asparagine, glutamine, glycine, and proline increased in MBO V2 (+32%, +211%, +75%, +52%, and +66%, respectively) and also in MBO V4 (+50%, +75%, +114%, +130%, and 102%, respectively), compared to controls (Figure 1A,B,F,G and Figure 2B,C). Also, glutamate increased only in MBO V2 (+28%) and serine only in MBO V4 (+38%), compared to control (Figure 1E and Figure 2C). In addition, proline increased in MBO V4 (+80%) (Figure 2B), threonine in MBO V2 (+420%) (Figure 2D), and glutamate in both vernalization procedures V2 and V4 (+109% and +32%, respectively) (Figure 1E) in the vegetative stage. Conversely, in this phase, glycine and serine decreased in MBO V2 (−66%, −72%, and −58%, respectively; Figure 1D,G and Figure 2G), asparagine (−24% and −67%; Figure 1B), and glutamine (−40% and −53%; Figure 1F) in both V2 and V4 compared to control. At the flowering stage, most amino acids showed a decrease in the V4 procedure, except asparagine in MBO V2, which showed an increase (+113%; Figure 1B). In particular, in V4, there was a decrease in alanine, glutamate, glutamine, glycine, MEA, proline, serine, and threonine contents (−95%, −86%, −99%, −66%, −32%, −81%, −96%, and −98%, respectively) (Figure 1A,E–H and Figure 2B–D), while GABA (−70% and −98; Figure 1D), and ornithine (−70% and −76; Figure 2A) decreased in V2 and V4 procedures, compared to controls. At flowering, alanine, asparagine, aspartate, GABA, glutamine, glutamate, and serine decreased only in MBO V2 (+87%, +99%, +98%, +93%, +99%, +90%, and +96%), while glycine decreased in both MBO V2 and V4 (−48% and −82%, respectively), compared to control.

The content of minor amino acids (including arginine, histidine, isoleucine, leucine, lysine, methionine, phenylalanine, tyrosine, tryptophan, and valine) was influenced by V and H × V (Appendix A). In fact, they increased in the pre-planting stage, in particular in both hybrids under V4 (+136% and 230%, respectively; Figure 2E). Minor amino acids also increased in the vegetative stage in MBO V2 and V4 (+48% and +51%, respectively) compared to controls. The branched-chain amino acids (BCAAs, including isoleucine, leucine, and valine) showed changes in all growth stages. BCAAs increased in pre-planting in MBO V4 (+291%) and in the vegetative phase in MDR V2 (+12%) compared to control (Figure 2F). On the contrary, these amino acids decreased during vegetative growth in MDR V4 (−40%) and only at the flowering stage in MBO V4 (−78%), compared to respective controls (Figure 2F).

The total amino acid content was mostly affected by V and H × V and less by H (Appendix A). Particularly, it was strongly influenced by V at pre-planting, as it decreased in MDR V2 (−14%) and increased in MBO V2 (+98%) and V4 (+84%), compared to controls (Figure 2G), and in the vegetative stage, as it decreased under both procedures in MDR (−65% and −71%, respectively) and only in MBO V4 (−61%) compared to respective controls. At flowering, there was a strong decrease (−95%) of the total amino acid content, but only in MBO V4 (−95%) compared to control (Figure 2G).

The soluble protein content was affected by H × V during vegetative growth and flowering (Appendix A). At vegetative stage, it was higher in MDR (+66%) than in MBO, and it increased in MDR V4 and MBO V2 compared to respective controls (+46% and +95%, respectively; Figure 2H). At flowering, soluble proteins increased in MDR V2 and V4 (+225% and 446%, respectively) compared to control, while they decreased in MBO V2 and V4 (−41% and −50%, respectively) compared to control. (Appendix A).

### 2.2. Principal Component Analysis (PCA)

A principal component analysis was performed on all metabolite data determined in tuberous root tissues of *Ranunculus asiaticus* L. hybrids MDR and MBO, subjected to the three vernalization procedures of pre-planting, vegetative growth, and flowering (Figure 3). The variables in the first four principal components (PCs) were highly correlated, with eigenvalues greater than 1, explaining 83.3% of the total variance, with PC1, PC2, PC3, and PC4 accounting for 46.5%, 22.1%, 8.7%, and 6.0%, respectively. PC1 was positively correlated to BCAAs, serine, alanine, glycine, GABA, threonine, glutamate, glutamine, and proline, whereas it was negatively correlated to polyphenols and aspartate. PC2 was positively correlated to MEA and asparagine, while it was negatively correlated to starch, fructose, and sucrose. The two hybrids in pre-planting were well separated along PC1, with MDR V2 and V4 close to the X axis on the positive side of PC1, while samples at flowering were on the negative side of PC1 and close to the axes intercept. The MDR samples in pre-planting were clustered on the negative side of PC2, while those in vegetative growth (MDR C) were clustered on the positive side of PC2.

## 3. Discussion

In this experiment, plants of the hybrids MBO (early flowering) and MDR (medium earliness) of *Ranunculus asiaticus* L. were subjected to three preparation procedures of tuberous roots and grown in an unheated glasshouse in the south of Italy from September to March. In our environmental and cultural conditions, flowering of control plants from only rehydrated roots started at the beginning of January, and the time for flowering was similar in the hybrids, despite the different earliness expected, and flower stem characteristics were comparable, as previously observed on the same genotypes subjected to the same preparation procedure [23].

Two-week vernalization procedures at 3.5 °C (V2) anticipated flowering in MDR (−27 days compared to control) and were ineffective in MBO, while those lasting 4 weeks (V4) promoted flowering in both hybrids (26 days in MBO and 58 days in MDR) but reduced the quality of flower stems, similarly to previous experiments on the same genotypes [14].

In Italy, *R. asiaticus* is mostly prepared through rehydration followed by vernalization, with planting from the end of August to the beginning of September and harvesting from the end of November to April. In different hybrids, the exposure of tuberous roots to 3.5 °C for 2 weeks after rehydration (as in our V2 treatment) anticipated flowering [1]. However, in our experiment, this procedure, which is the most common in breeding farms, was effective in MDR but ineffective or detrimental in MBO, confirming that cold requirements are hybrid-specific [1,24]. Vernalization reduced the flower fresh weight in both the hybrids and also the stem height in MDR, with a stronger effect after 4 weeks of treatment as observed in a previous experiment. Ohkawa investigated the influence of two vernalization procedures, 5 °C for 2 weeks and for 4 weeks, compared to only rehydration, in 2 cultivars of *R. asiaticus* and found a positive relationship between flowering anticipation and the duration of cold treatment in one genotype (as we observed in MDR) and a negative influence of the longer treatment on flowering time and quality in the other [24].

The targeted metabolic profiling in MDR and MBO tuberous roots, after the three treatments and in the three different developmental phases, showed a significant interaction between genotype and vernalization procedure in relation to the content of amino acids, soluble sugars, and polyphenols, as previously reported by Fusco et al. [18] in leaves. The first and main change at the pre-planting phase in the two hybrids, under V2 and V4 treatments, was the stronger increase in γ-aminobutyric acid (GABA), alanine, and proline in MDR compared to MBO, even if the absolute values of these amino acids were higher in MBO than in MDR. Moreover, MDR, unlike MBO, did not show a further consistent increase of all other amino acids or metabolites in the root tissues previously vernalized. Indeed, the strong and fast increase of GABA, proline, and alanine together is a clear symptom of stress for the tuberous root. Vernalization-related cold stress may cause dysregulation of pH, oxidative stress, and metabolic disfunction [25], particularly in plants sensitive to cold stress, as already proven for MDR [18]. The increase of GABA, alanine, and proline in the pre-planting phase may alleviate the effects of cold stress by buffering the cytoplasmic acidosis and potentiating the antioxidant defense [21,26]. In fact, the synthesis of both alanine and GABA involves proton-consuming reactions, thus regulating cytosolic pH. In particular, alanine is synthetized through decarboxylation of malate to pyruvate by malic enzyme activity and then transamination of pyruvate to alanine, while GABA is obtained through glutamate decarboxylation catalyzed by glutamate decarboxylase [21,27,28]. In its turn, proline is not only well known for its ability to act as an osmolyte, but it may also detoxify ROS, buffer cellular redox potential, and protect cellular structures from oxidative stress [20]. In addition, GABA is also able to scavenge ROS (e.g., hydrogen peroxide, singlet oxygen, and superoxide anion radicals) like or even better than proline at similar concentrations [29,30]. Molina-Rueda et al. [30] reported that GABA and proline are often synthesized and accumulated at the same time in response to environmental stresses to protect plant tissues from oxidative stress. Finally, after relief from cold stress, alanine, GABA, and proline can be metabolized and/or converted to intermediates of the Krebs cycle and used to produce alpha-keto acids and ATP, thus re-activating plant metabolism and accelerating flowering [26,31]. Probably the initial higher capacity to synthesize and accumulate alanine and GABA, in addition to branched chain amino acids (BCAAs), in control conditions preserved MBO from the symptoms of cold stress determining its lower sensitivity to shorter-term cold stress (V2 treatment). It has previously been suggested [14] that BCAAs in ranunculus may work as ROS scavengers, reducing the onset of oxidative damage, in addition to acting as alternative electron donors for the mitochondrial electron transport chain. However, under V4 treatment, MBO started having more major problems coping with cold-related stress due to the vernalization procedure than MDR and therefore accumulated larger amounts of glycine. It has been proven that accumulation of endogenous glycine or supply of exogenous one may mitigate plant stresses (e.g., cold stress), even if it can reduce the development of plants, as seen in the previous studies [32]. Glycine can then be converted into glutamate and decarboxylated to GABA, which, by means of a GABA shunt, can have an anaplerotic function [33].

During vegetative growth, the strong increase of asparagine in the tuberous roots of MDR C is possibly due not only to the hydrolysis of root proteins but also to the import of asparagine from leaves. Asparagine, in fact, plays a key role in nitrogen transport because of its N:C ratio of 2:4 and can be rapidly deaminated, producing aspartate and then oxaloacetic acid that can be used in the citric acid cycle or for gluconeogenesis. Carbon skeletons and ATP may in fact be used to boost the growth of tuberous roots in the vegetative phase, thus delaying the growth of stems and leaves and therefore flowering.

During the vegetative and flowering phases, the content of polyphenols also increased. Zhou et al. [34] found that tobacco plants subjected to 4 °C, similarly to *Ranunculus*, showed an increase in the content of various metabolites that fall into the class of polyphenols. This was considered a positive self-protection mechanism, able to increase the strength and rigidity of the cell wall and the water transport in the vascular system, which were compromised by low temperatures. However, this is a high-energy-consuming process that diverts intermediates and energy from growth, thus causing lower root and stem dry weight, thinner and shorter stems, and early flowering. Also, the increase of MEA in MDR V4 and MBO V2 and V4, obtained for decarboxylation of serine, may play an important role in the synthesis and/or regeneration of phospholipids [14].

Differently from MDR, MBO started to accumulate free sugars, starch, and proteins in tuberous roots already during the vegetative stage, particularly under V2 and V4 treatments, showing that in this hybrid the shorter vernalization activated the storage of surplus reserves even before flowering without affecting the stem dimension and/or the time of flowering. In MBO V4, at the vegetative stage, high amounts of fructose and sucrose were synthetised. Sucrose is a storage carbohydrate that can be rapidly mobilized according to metabolic needs and rapidly loaded into the translocation stream [35]. In fact, it was probably exported to leaves where it could anticipate flowering; in fact, the increase in sucrose is correlated with flowering in several species [36]. Furthermore, this metabolite may also be a compatible compound able to stabilize cell membranes and maintain turgidity [37].

## 4. Materials and Methods

The experiment was carried out at the Department of Agriculture of the University of Naples (Portici, Italy—40°49′ N, 14°20′ E) from 18 September 2018 to 30 March 2019.

Dry tuberous roots of two hybrids of *Ranunculus asiaticus* L., MBO (early flowering, pale orange flower) and MDR (medium earliness, bright orange flower) (Biancheri Creazioni, Italy, https://www.bianchericreazioni.it/ accessed on 31 July 2023) were subjected to three preparation procedures:-Only rehydration: exposure to water spraying at 12 °C for 24 h in a humid chamber after soaking in tap water, consisting of a subsequent cycle of immersion spaced out by drainage to prevent anoxia (Control, C);-Rehydration followed by vernalization at 3.5 °C for 2 weeks (V2);-Rehydration followed by vernalization at 3.5 °C for 4 weeks (V4).

Tuberous roots of the most common size for each hybrid were used (3–4 cm for MBO and 4–5 cm for MDR). Plants were grown in pots on a mixture of perlite and peat (70:30 in vol.). Irrigation was alternated with fertigation (4 pulses per week in total). In the Hoagland full-strength nutrient solution, pH and electrical conductivity (EC) were kept at 5.5 and 1.7 dS/m, respectively, and monitored with a portable pH-EC sensor (HI9813 series, Hanna Instruments Intl.). The mean values of air temperature and relative humidity (day/night) recorded during the experiment were 23.7 ± 5.0/12.30 ± 4.1 °C and 58.5 ± 6.8/74.3 ± 16.9%, respectively (Mean Value ± Standard Deviation).

### 4.1. Metabolic Analyses

For the pre-planting phase, tuberous root samples were randomly selected within the propagation material provided by the breeder (Biancheri Creazioni, Italy, https://www.bianchericreazioni.it/ accessed on 31 July 2023). For the phases of vegetative growth and flowering, 3 plants were randomly collected for each combination Hybrid x Propagation procedure.

#### 4.1.1. Starch and Soluble Carbohydrate Analysis

Starch and soluble sugars were determined according to Fusco et al. [18], with some modifications. 10 mg of lyophilized tuberous roots were suspended twice in 140 µL ethanol 80% (*v*:*v*) and once in 70 µL ethanol 50% (*v*:*v*) at 80 °C in a thermomixer (Eppendorf ThermoMixer^®^ C) for 20 min, and then centrifuged at 14,000 rpm for 10 min at 4 °C. The clear supernatants, separated from pellets, were combined and stored in 1.5 mL tubes at −20 °C until analysis of glucose, fructose, and sucrose. The pellets of ethanolic extraction were extracted with 500 μL of 0.1 M KOH, heated at 90 °C for 2 h [38], cooled in ice, and then acidified to pH 4.5 with acetic acid. An aliquot of acidified samples was mixed with sodium acetate 50 mM pH 4.8, α-amylase 2 U/mL and amyloglucosidase 20 U/mL and incubated at 37 °C for 18 h for enzymatic hydrolysis. The samples were centrifuged at 14,000 rpm for 10 min at 4 °C, and the supernatant containing the glucose derived from starch hydrolysis was used for measurement. The content of soluble sugars in the ethanolic extracts and the glucose derived by starch hydrolysis were determined by an enzymatic assay coupled with a reduction in pyridine nucleotides, and the increase in absorbance at 340 nm was recorded using a Synergy HT spectrophotometer (BioTEK Instruments, Bad Friedrichshall, Germany). The amount of NADH formed in the reaction was stoichiometric to the amount of glucose in the sample. The absorbance of samples was referred to the calibration curve of glucose standards, and the content of sugars was expressed as mg g^−1^ DW.

#### 4.1.2. Soluble Proteins, Free Amino Acid Analysis

Soluble proteins (mg g^−1^ DW) were extracted from 10 mg of lyophilized tuberous root samples with a buffer containing 200 mM TRIS-HCl pH 7.5 and 500 mM MgCl_2_. Then the samples were vortexed and stored at 4 °C for 24 h. After 24 h, the samples were centrifuged at 14,000 rpm for 10 min at 4 °C. The clear supernatants (10 µL) were added to 190 µL of Bio-Rad protein assay dye reagent diluted 1:5 (*v*:*v*) with bidistilled water [39]. The solutions were mixed, and then absorbance at 595 nm was recorded on a microplate reader (Synergy HT, BioTEK Instruments, Bad Friedrichshall, Germany). The soluble protein content in the samples was calculated by comparison with standard curves obtained using known concentrations of bovine serum albumin (BSA) as the reference standard, as reported in Carillo et al. [40]. Free amino acids and proline were determined by the method described by Woodrow et al. [20]. Aliquots of about 10 mg of lyophilized samples were extracted in 1 mL of ethanol:water (40:60 *v*:*v*) overnight at 4 °C. After 24 h, the samples were centrifuged at 14,000 rpm at 4 °C for 10 min. Free amino acids were determined by high-performance liquid chromatography (HPLC) with a fluorescent detector (FLD) after precolumn derivatization with ophthaldialdehyde (OPA) according to the method described in Carillo et al. [41]. The HPLC peaks were identified and quantified by comparing their retention times and area values with those of amino acid standards (HPLC-grade) (Appendix A). Amino acid standard solutions (A9906 Fluka, Sigma-Aldrich—Merck, St. Louis, USA) were combined 1:1 (*v:v*) with ethanol:water in the ratio 40:60 (*v*:*v*) and derivatized with OPA reagent (1:2, *v*:*v*) for 3 min in the autosampler needle, injected onto the column, and eluted as the sample. Proline was determined in the same ethanolic extract used for the amino acids’ determination by an acid ninhydrin colorimetric method, according to Woodrow et al. [20]. The absorbance of samples was referred to as the calibration curve of proline standards. The amino acids were expressed as µmol g^−1^ DW. 

#### 4.1.3. Polyphenols Analysis

Polyphenols (mg GAE g^−1^ DW) were determined according to Singleton et al. [42] with some modifications. 30 mg of lyophilized tuberous root samples were extracted in 700 μL of 60% methanol (*v*:*v*) and vortexed. Then the samples were centrifuged at 1000 rpm for 10 min at 25 °C. An aliquot of clear supernatant (35 μL) was mixed with 125 μL of the Folin–Ciocalteu reagent, diluted 1:4 (*v*:*v*) with bidistilled water, and mixed for 6 min at room temperature. After 6 min, 650 μL of 3% (*v*:*v*) sodium carbonate was added. After 90 min at room temperature, the absorbance at 760 nm was determined in a microplate reader (Synergy HT, BioTEK Instruments, Bad Friedrichshall, Germany). Polyphenol concentrations were determined against standard curves of gallic acid as described in Carillo et al. [40] and expressed as mg GAE g^−1^ DW.

## 5. Statistical Analysis

The experiment was conducted on 25 plants per combination of “*Hybrid × Vernalization procedures*.” All the other analyses were performed on three biological replicates for each treatment/hybrid. Data were subjected to statistical analysis by using SigmaPlot 12.0 (SPSS Inc., Norman Nie Dale Bent, Hadlai “Tex” Hull, Chicago, IL, USA) software package. The main effect of the categorical independent factors (i.e., hybrids and preparation procedures) and their interaction on the continuous dependent variables were analyzed through a two-way ANOVA. In the case of rejection of the null hypothesis, the Tukey’s HSD test was performed (*p* ≤ 0.05). 

## 6. Conclusions

In conclusion, the two hybrids showed different responses to the root preparation procedure. The MBO hybrid, being able to synthesize higher constitutive amounts of protecting metabolites like GABA and alanine, better overcame the negative effects of vernalization, at least when applied for a shorter time (V2). On the contrary, the MDR hybrid was more sensitive to vernalization, decreasing its capacity to export nutrients to leaves due to the need to rapidly synthesize high-energy-cost organic osmolytes for coping with dysregulation of pH, oxidative stress, and metabolic disfunction caused by cold stress. Moreover, the MDR needs to synthesize continuously protecting metabolites, like polyphenols, also in roots during the vegetative stage, contributing to subtract carbon intermediates and ATP from growth to contrast the oxidative effects of vernalization and the consequent metabolic imbalance. The new knowledge about the metabolic responses of the hybrids to vernalization may be directly translated into the commercial field and used for implementing the best hybrid-specific preparation procedures at the breeding farm level.

## Figures and Tables

**Figure 1 plants-12-03255-f001:**
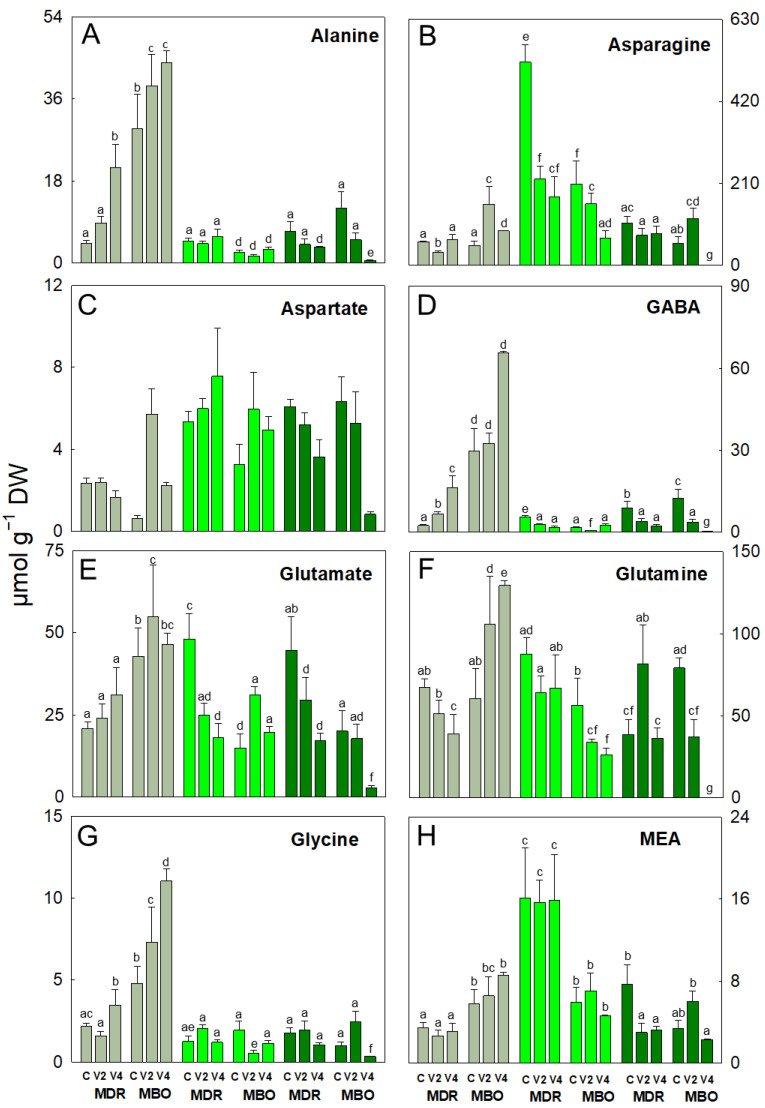
Alanine (**A**), asparagine (**B**), aspartate (**C**), γ-aminobutyric acid (GABA, (**D**)), glutamate (**E**), glutamine (**F**), glycine (**G**), and monoethanolamine (MEA, (**H**)) expressed as µmol g ^−1^ DW in plants of *Ranunculus asiaticus* L. hybrids MDR and MBO, obtained by three vernalization procedures of tuberous roots: only rehydration (Control, (**C**)), rehydration plus vernalization for 2 weeks (V2), and rehydration plus vernalization for 4 weeks (V4), throughout the growing cycle, in different plant stages, pre-planting (greyish), vegetative phase (light green), and flowering (dark green). All data are expressed as mean ± SD, *n* = 3. Different letters indicate significant differences (*p* ≤ 0.05).

**Figure 2 plants-12-03255-f002:**
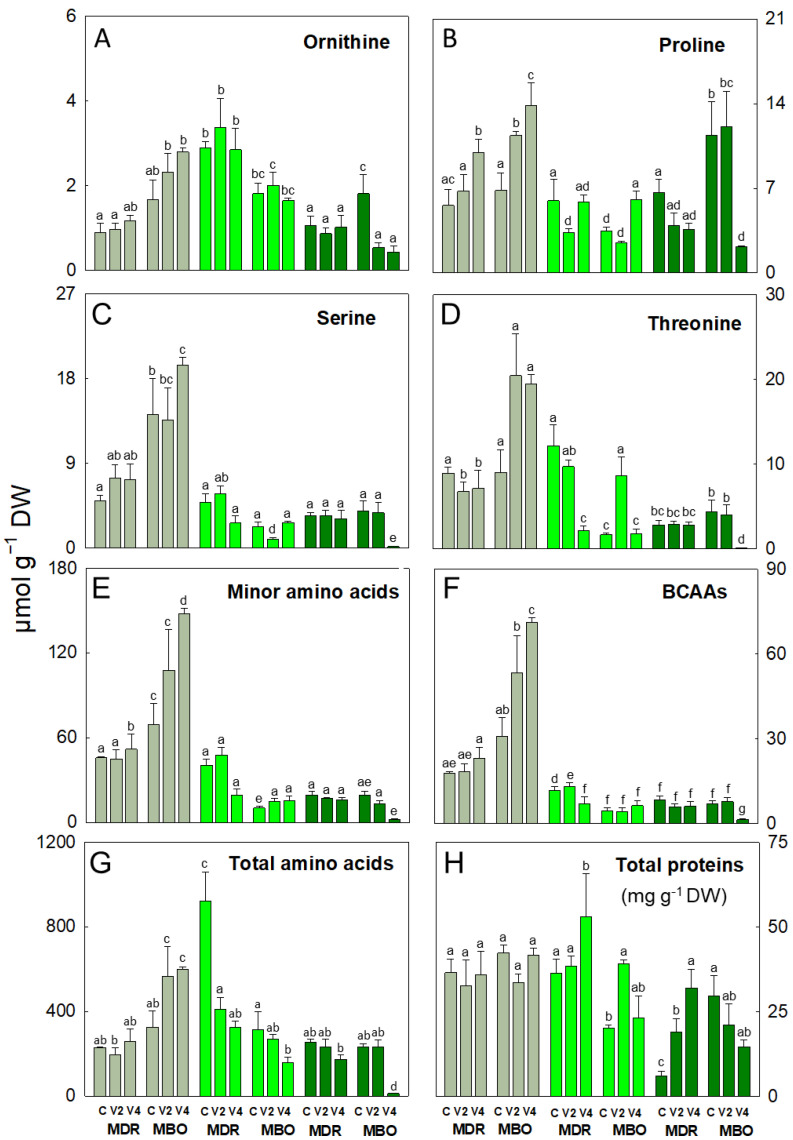
Ornithine (**A**), proline (**B**), serine (**C**), threonine (**D**), minor amino acids (**E**), branched chain amino acids (BCAAs, (**F**)), and total amino acids (**G**) expressed as µmol g ^−1^ DW, and total soluble proteins (**H**) expressed as mg g^−1^ DW, in plants of *Ranunculus asiaticus* L. hybrids MDR and MBO, obtained by three vernalization procedures of tuberous roots: only rehydration (Control, (**C**)), rehydration plus vernalization for 2 weeks (V2), and rehydration plus vernalization for 4 weeks (V4), throughout the growing cycle, in different plant stages, pre-planting (greyish), vegetative phase (light green), and flowering (dark green). All data are expressed as mean ± SD, *n* = 3. Different letters indicate significant differences (*p* ≤ 0.05).

**Figure 3 plants-12-03255-f003:**
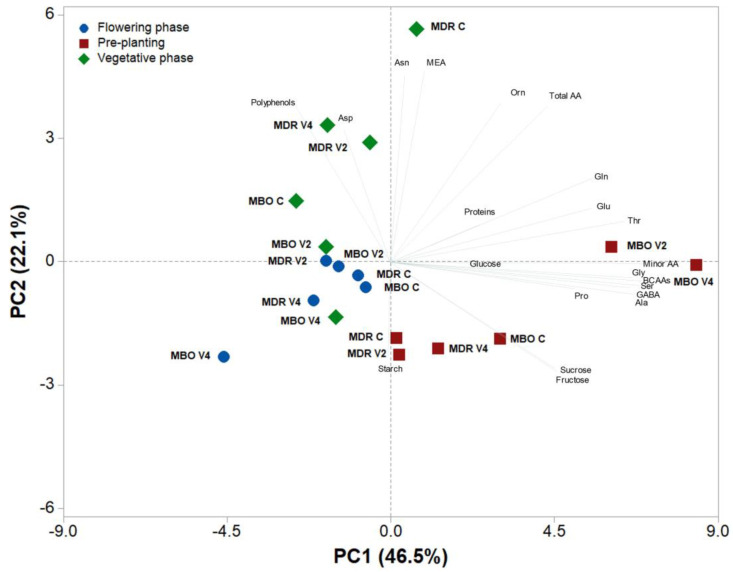
Principal component loading plot and scores of principal component analysis (PCA) of carbohydrates, amino acids, polyphenols, and soluble proteins in roots of *Ranunculus asiaticus* L. hybrids MDR and MBO, obtained by three vernalization procedures of tuberous roots: only rehydration (Control, C), rehydration plus vernalization for 2 weeks (V2), and rehydration plus vernalization for 4 weeks (V4), throughout the growing cycle, in different plant stages (pre-planting, vegetative phase, and flowering).

**Table 1 plants-12-03255-t001:** Polyphenols (in µg g^−1^ DW), glucose, fructose, sucrose, and starch (in mg g^−1^ DW), in plants of *Ranunculus asiaticus* L. hybrids MDR and MBO, obtained by three vernalization procedures of tuberous roots: only rehydration (Control, C), rehydration plus vernalization for 2 weeks (V2), and rehydration plus vernalization for 4 weeks (V4), throughout the growing cycle, in the different plant stages (pre-planting, vegetative phase, and flowering). ns, *, **, and ***; indicate a non-significant or significant difference at *p* ≤ 0.05, *p* ≤ 0.01, and *p* ≤ 0.001, respectively. Different lowercase or capital letters within each row, for a specific vernalization procedure, indicate significant differences at *p* ≤ 0.05.

		MDR			MBO				H	V	H × V
	C	V2	V4	*Mean*	C	V2	V4	*Mean*			
*Pre-planting phase*											
Polyphenols	4.05 ab	3.30 a	4.28 b	*3.88*	3.67 ab	3.61 a	3.92 b	*3.73*	ns	*	ns
Glucose	24.96 ab	24.01 bc	27.52 a	*25.5 A*	22.95 bd	21.22 cd	19.94 d	*21.37 B*	***	ns	**
Fructose	23.50	20.94	23.36	*22.60*	21.69	21.76	21.09	*21.51*	ns	ns	ns
Sucrose	8.78	8.35	8.76	*8.63 A*	8.41	8.13	8.30	*8.28 B*	*	ns	ns
Starch	41.05 a	38.38 a	39.21 a	*39.55 B*	54.14 bc	59.82 b	44.61 ac	*52.86 A*	***	ns	**
*Vegetative phase*											
Polyphenols	13.97 a	16.00 b	7.94 cd	*12.64 A*	7.64 cd	11.21 c	5.16 d	*8.01 B*	*	**	**
Glucose	20.76 a	21.47 a	10.25 b	*17.49*	18.97 a	8.80 b	22.47 a	*16.75*	ns	ns	***
Fructose	7.45 a	6.17 a	11.58 b	*8.40*	6.62 a	2.27 c	22.50 d	*10.46*	ns	**	***
Sucrose	2.86 a	2.15 a	4.43 b	*3.15*	2.28 a	0.52 c	8.46 d	*3.76*	ns	**	***
Starch	27.18 ab	28.64 ab	89.43 a	*48.41*	27.42 ab	20.95 b	35.46 a	*27.94*	ns	**	*
*Flowering phase*											
Polyphenols	5.85 a	12.85 b	11.69 b	*10.13 A*	10.82 b	4.77 ac	2.45 c	*6.01*	*	ns	***
Glucose	35.63 a	31.04 a	18.59 bc	*28.42 A*	15.26 c	23.58 b	1.08 d	*13.31 B*	**	*	***
Fructose	13.49 a	13.89 a	12.72 a	*13.36*	10.63 a	13.13 a	0.42 b	*8.06*	ns	ns	***
Sucrose	5.79 a	6.15 a	4.78 a	*5.57 A*	2.12 b	3.07 b	0.09 c	*1.76 B*	***	ns	***
Starch	40.24 acd	49.75 ab	25.28 c	*38.42 B*	47.61 bd	60.79 b	52.83 b	*53.75 A*	*	ns	*

## Data Availability

All the data are reported within the article.

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
