# Peer review of "Metabolic Profiling in Tuberous Roots of Ranunculus asiaticus L. as Influenced by Vernalization Procedure"

_plants, 2023, doi:10.3390/plants12183255_

Round 1

Reviewer 1 Report (Previous Reviewer 1)

I have completed the review of the manuscript under reference. Authors have investigated changes in metabolites in two hybrids of R. asiaticus, an ornamental plant. I have several concerns, as detailed hereunder:

1. What was the rationale for undertaking the study?

2. Why two hybrids were chosen for the study?

3. Why the metabolites were analysed?

4. What was the relation to vernalization treatments? I could not find the hypothesis for the presented work.

5. Methods need to be explained in an elaborated manner.

5.  In addition, the manuscript needs extensive revision as far as the presentation and English grammar is concerned. 

The ms is full of grammatical errors, and plurality issues. At several places, there are syntax errors and phrases have been used inappropriately. Sentence structuring is also poor. For example, studied the influence of two hybrids ..

Author Response

We thank the reviewer for his comments and suggestions. We addressed in the manuscripts all the requests. Below our response to his questions.

RC = reviewer comment

OR = Our response

I have several concerns, as detailed hereunder:

RC = 1. What was the rationale for undertaking the study?

OR = Due to the importance of tuberous roots as both plant reserve organ and propagation material, new knowledge on their metabolism can help understanding the source-sink relationships with the plant aerial part in the different developmental stages. Within a series of experiments aiming at elucidating the influence of several factors (i.e., temperature and photoperiod), and their interaction, on plant physiology, metabolism, and flowering of R. asiaticus, we investigated the effects of three preparation procedures of tuberous roots on plant growth and metabolic profile in two hybrids (differing in flowering earliness), grown in unheated glasshouse. Our hypothesis was that monitoring quantitatively specific metabolites (i.e., carbohydrates and amino acids, polyphenols), synthesized by plants in response to cold/oxidative stress for their protective function, could help highlighting the diverse cold requirements, hence the different genotype sensitivity to vernalization procedures (as demonstrated in the paper). This targeted metabolic profiling in both storage organ and leaves contributes at unveiling the strategies adopted by the hybrids to overcome the cold/oxidative stress deriving from vernalization and is directly transferable in field for flowering scheduling of ranunculus.

RC = 2. Why two hybrids were chosen for the study?

OR = As cultivation of Ranunculus asiaticus L. has been rising during the last years all over the world, breeders have been developing a relevant number of new hybrids, with different physiological and morphological features, depending on the diverse climatic areas and local market demands. We carried out a series of experiments aiming at elucidating the plant metabolic responses to several treatments to control the flowering time and characteristics (i.e., vernalization of tuberous roots, photoperiodic lighting in greenhouse). These experiments intended to acquire basic knowledge on plant metabolism as well as useful information for practical management of propagation and cultivation, to be transferred to commercial farms. In this respect, to match the industrial needs, the experimental treatments were defined in cooperation with Biancheri Creazioni - Italy (https://www.bianchericreazioni.it), a leader company of breeding in Ranunculus, and the choice of the hybrids was driven by experts and led to the two hybrids, as representative of genotypes with potential different responses, since they show different time for flowering (MBO, early flower) and MDR (medium earliness).

RC = 3. Why the metabolites were analysed?

OR = We analysed the selected metabolites because they are quantitively determinable and directly related to plant growth.

RC = 4. What was the relation to vernalization treatments? I could not find the hypothesis for the presented work.

OR = As mentioned above, we hypothesized that the quantification of specific metabolites (i.e., carbohydrates and amino acids, polyphenols), synthesized by plants in response to cold/oxidative stress, could help highlighting the diverse genotype sensitivity to vernalization procedures, hence the choice of the best preparation procedure to schedule flower production of ranunculus by the breeder.

RC = 5. Methods need to be explained in an elaborated manner.

OR = We decided to avoid detailed description of analytical procedures because they are clearly explained in other papers (also published in the same journal). Repeating this explanation would have implied self-plagiarism.

RC = 5. In addition, the manuscript needs extensive revision as far as the presentation and English grammar is concerned.

OR = We thank the Reviewer for highlighting the poor writing of some paragraphs, which was due to several reasons (i.e., overlapping of numerous deadlines before the summer break and insufficient control of the contribution of young researchers). The manuscript has been reviewed by a English mother tongue, many errors have been corrected, and several parts have been rephrased.

RC = The ms is full of grammatical errors, and plurality issues. At several places, there are syntax errors and phrases have been used inappropriately. Sentence structuring is also poor. For example, studied the influence of two hybrids.

OR = We thank the Reviewer for highlighting this weakness. We apologize for the poor writing of some paragraphs, due to several reasons (i.e., overlapping of numerous deadlines before the summer break and insufficient control of the contribution of young researchers). The whole manuscript has been reviewed by a English mother tongue, the grammatical and syntax errors have been corrected, and several parts have been rephrased.

Reviewer 2 Report (Previous Reviewer 3)

This paper was significantly improved. A few minor comments are presented in the attached file.

Author Response

We thank the reviewer for his comments and suggestions. We addressed in the manuscripts all the requests. Below our response to his questions.

RC = reviewer comment

OR = Our response

RC = This paper was significantly improved. A few minor comments are presented in the attached file.

OR = We thank the Reviewer for providing these last comments. All the suggested corrections have been included in the current version of the manuscript. Referring the suggestion to clarify the verb “to anticipate”, referred to the flowering process, we decided do not change it, to maintain the uniformity with our previous papers as well as with a relevant part of the literature on the topic. Scheduling of plant production is a critical aspect in modern floriculture, since nowadays sales of cut flowers and ornamentals are not oriented to recurring holidays as in the past, but more to impulse buying, implying a more diverse and constant demand on the market. This requires a continuous production, hence diverse techniques to modulate the duration of the growing cycle, by hastening or slowing the plant growth and development, have been developed. Among the numerous approaches, manipulation of climatic parameters (e.g., temperature before or after planting) in the growth environment is one of the most common in greenhouse floriculture and the control of the storing temperature of bulbs is the most effective in geophytes with only one flowering. On this basis, the efficiency of the strategies is often expressed in terms of their ability to promote flowering and “to anticipate” is a common term to describe this effect (see Proietti S., Scariot V., De Pascale S., Paradiso R., 2022. Flowering mechanisms and environmental stimuli for flower transition: bases for strategies of production scheduling in greenhouse floriculture. Review article. Plants, 11, 432. https://doi.org/10.3390/plants11030432).

Reviewer 3 Report (Previous Reviewer 4)

I have read the attached files with email 

Everything seems to be good except my previous request to provide the outputs of HPLC at least as a supplementary with this manuscript. This  is extremely important for the credibility of your work

Author Response

We thank the reviewer for his comments and suggestions. Below our response to his last request.

RC = reviewer comment

OR = Our response

RC = Everything seems to be good except my previous request to provide the outputs of HPLC at least as a supplementary with this manuscript. This is extremely important for the credibility of your work.

OR = A selection of the amino acids chromatograms have been now included as examples as supplementary figures. In addition, an accurate revision of English has been made by a mother tongue proof-reader.

Round 2

Reviewer 1 Report (Previous Reviewer 1)

The manuscript has been considerably improved. I have few minor suggestions:

Authors should specify what was the difference in two cultivars? It should be explained in abstract too!

No use of abbreviations in abstract pl.

It is largely ok. 

Author Response

We thank the reviewer for this last suggestion.

The flowering earliness and the flower color have been added and the abbreviations have been removed in the Abstract. The flower color has been added in Materials and Methods.

This manuscript is a resubmission of an earlier submission. The following is a list of the peer review reports and author responses from that submission.

Round 1

Reviewer 1 Report

Authors have investigated the potential role of vernalization in influencing growth and metabolism The presented work is worthy of publication. However, I have a few concerns. These are:

What was the rationale for the work? Why the study was undertaken? A robust hypothesis is missing!

The manuscript is purely a descriptive one and does not provide a mechanistic explanation/ hypothesis. 

Why only amino acids, phenols and carbohydrates were analysed? What about other metabolites? 

What new information is presented?

Another concern is that manuscript lacks a conclusion that summarizes the main findings of the study?

Author Response

RC = reviewer comment

OR = Our response

RQ = What was the rationale for the work? Why the study was undertaken? A robust hypothesis is missing!

The manuscript is purely a descriptive one and does not provide a mechanistic explanation/ hypothesis.

OR = We thank the reviewer for highlighting this weakness. Our hypothesis has been clarified in the current version (page 3, lines 469-472).

RQ = Why only amino acids, phenols and carbohydrates were analysed? What about other metabolites? What new information is presented?

OR = This aspect has been clarified in the revised version of the manuscript (page 3, lines 461-469).

RQ = Another concern is that manuscript lacks a conclusion that summarizes the main findings of the study?

OR = As required by the Author’s guidelines and reported in the manuscript template for Plants journal, the section “Conclusions” is placed at the end of the manuscript, after Materials and Methods (paragraph 6). It is possible that the Reviewer missed the paragraph since it is not in the typical position, after the Discussion.

Reviewer 2 Report

Dear authors,

Could you please explain the following:

  1. How did you obtain concentrations of amino acids? Did you use an internal standard? Please provide the procedure used.

  2. Why did you not perform biological replicates for each entity (C, H, V)?

  3. How can this newly acquired knowledge be applied?

Best regards,

Author Response

RC = reviewer comment

OR = Our response

Could you please explain RC = reviewer comment

OR = Our response

the following:

RC = How did you obtain concentrations of amino acids? Did you use an internal standard? Please provide the procedure used.

OR = In the Materials and methods section has been added “The HPLC peaks were identified and quantified comparing their retention times and area values with those of amino acid standards (HPLC-grade). Amino acid standard solutions (A9906 Fluka, Sigma-Aldrich) were combined 1:1 (v.v) with ethanol:water in the ratio 40:60 (v/v) and derivatized with OPA reagent (1:2, v:v) for 3 min in the autosampler needle, injected onto the column and eluted as the sample.”

RC = Why did you not perform biological replicates for each entity (C, H, V)?

OR = We performed three biological replicates for each treatment/hybrid. We have now specified it in the section “statistical analysis”.

RC = How can this newly acquired knowledge be applied?

OR = The research was carried out within a series of experiments aiming at investigating the plant metabolism in Ranunculus asiaticus L. with the final objective to improve the propagation technique, which must be able to allow breeders to differentiate the vernalization procedure depending on the specific plant genotype and the final destination of propagation materials in terms of geographic areas. Potential applications of our data are related to the possibility to exploit the different plant metabolic response to cold temperature to better foresee and schedule flowering in commercial farms. Further studies are in progress to achieve this objective.

Reviewer 3 Report

1. Some parts are taken from the previous publication of the same authors in Plants. Although the wording was slightly changed, it looks like self-plagiarism and has to be completely avoided. As an example, in the section Materials and Methods sampling of the aerial part is described, while the paper is dedicated to the underground organs. 2. I am not sure that the authors are familiar with the annual cycle of Ranunculus roots. These are annual organs that are completely replaced during vegetation and flowering, therefore they probably measured different structures after planting and prior to harvest. This is not clarified. 3. The results on the aerial parts from the previous paper are repeated. It might be useful for the discussion, but section 2.1 has to be reduced Please see attached file.

Please carefully review your manuscript to avoid any errors in grammar.

Author Response

RC = reviewer comment

OR = Our response

RC = Some parts are taken from the previous publication of the same authors in Plants. Although the wording was slightly changed, it looks like self-plagiarism and has to be completely avoided. As an example, in the section Materials and Methods sampling of the aerial part is described, while the paper is dedicated to the underground organs.

OR = We thank the reviewer for highlighting this excessive overlapping. Some paragraphs have been completely removed and some have been changed to avoid it in the current version. Materials and methods section has been also improved by detailing the use of tuberous root samples, standards and units of measure.

RC = I am not sure that the authors are familiar with the annual cycle of Ranunculus roots. These are annual organs that are completely replaced during vegetation and flowering, therefore they probably measured different structures after planting and prior to harvest.

OR = The annual life cycle of tuberous roots has been clarified (page 2, lines 62-64). In addition, as suggested, we have enriched the manuscript reviewing recent articles dealing with starch breakdown after vernalization in potato and gladioulus.

RC = The results on the aerial parts from the previous paper are repeated. It might be useful for the discussion, but section 2.1 has to be reduced. Please see attached file.

OR = Data on plant growth have been removed in the Results and are now cited only in Discussion, and the paper is duly cited.

Reviewer 4 Report

In the present study, the authors investigated the influence of three preparation procedures of tuberous roots in Ranunculus asiaticus L, including only rehydration (control, C) and two times (2, 4 weeks; V2 & V4) of exposure to cold temperature (3.5 °C) after rehydration, on plant growth, and metabolic profile in two hybrids (MBO and MDR ) with different flowering earliness. The metabolic profile was observed in different plant stages (pre-planting, vegetative phase and flowering). Furthermore, the authors demonstrated the importance of the existence of metabolites e.g., amino acids in particular the branched chain amino acids (BCAAs) as protective agents against the adverse effects of low temperatures.

The authors found that the two hybrids responded differently to the root preparation process. The MBO hybrid was better able to withstand the detrimental effects of vernalization, at least when it was applied for a shorter period of time (V2), due to the presence of GABA and alanine. The MDR hybrid, on the other hand, was shown to be more sensitive to vernalization, reducing its ability to export nutrients to leaves in order to compete with the dysregulation of pH, oxidative stress, and metabolic dysfunction brought on by cold stress. Furthermore, it has been found that the MDR produced polyphenols in roots during the vegetative stage.

The manuscript is prepared well and written in good English; however, i have some suggestions which may improve the overall quality of this manuscript before the final publication

1- The reference in the abstract section should be removed (it is unusual to write a reference in this section) just refer to treatments, results, conclusion and recommendation  

2- There is no relevant experimental research on molecular level

3- The authors didn’t measure antioxidant enzyme activity in the paper under low temperature. As well as the transcript levels of antioxidant enzyme genes should be measured to support the experiments and the discussion section

4- The references section should be more supported.  There is a shortage in references generally and the recent references specifically

5- The results of HPLC should be provided as figures at least in the supplementary

Author Response

RC = reviewer comment

OR = Our response

The manuscript is prepared well and written in good English; however, I have some suggestions which may improve the overall quality of this manuscript before the final publication

RC = The reference in the abstract section should be removed (it is unusual to write a reference in this section) just refer to treatments, results, conclusion and recommendation.

OR = We thank the reviewer for highlighting this oversight. The reference in the Abstract has been removed.

RC = There is no relevant experimental research on molecular level

OR = In this MS we wanted to show the metabolic changes mainly of primary metabolites able to be used for coping with cold stress due to vernalization and promptly recyclable to address plant growth requirements. Moreover, in this experiment, lyophilized plant material was analyzed for metabolic profiling. For molecular analyses, samples must be shock frozen in liquid nitrogen and stored at -80°C until molecular analysis. We thank the reviewer for the suggestion, and we will perform a new research focusing on this aspect.

RC = The authors didn’t measure antioxidant enzyme activity in the paper under low temperature. As well as the transcript levels of antioxidant enzyme genes should be measured to support the experiments and the discussion section

OR = For enzymatic analyses, samples must be shock frozen in liquid nitrogen and stored at -80°C until analysis, too. We will perform the new research focusing on both molecular and enzymatic aspect.

RC = The references section should be more supported. There is a shortage in references generally and the recent references specifically

OR = The literature cited has been integrated.

RC = The results of HPLC should be provided as figures at least in the supplementary

OR = The data of amino acids and soluble proteins are now shown by Figures 1 and 2 in the article, in addition to a Supplemental Table with data and statistical analysis.
